# DRIP: Decompositional reasoning for Robust and Iterative Planning with LLM Agent

## Abstract

Research on LLM agents has shown remarkable progress, particularly in planning methods that leverage the reasoning capabilities of LLMs. However, challenges such as robustness and efficiency remain in LLM-based planning, with robustness, in particular, posing a significant barrier to real-world applications. In this study, we propose a framework that incorporates human reasoning abilities into planning. Specifically, this framework mimics the human ability to break down complex problems into simpler problems, enabling the decomposition of complex tasks into preconditions and subsequently deriving subtasks. The results of our evaluation experiments demonstrated that this human-like capability can be effectively applied to planning. Furthermore, the proposed framework exhibited superior robustness, offering new perspectives for LLM-based planning methods.

## 1 Introduction

The evolution of Large Language Models (LLMs) has been remarkable, extending their influence to interdisciplinary domains. Among these advancements, the emergence of LLM-powered agent technology (LLM Agents) has garnered significant attention due to its potential for real-world applications. These agents leverage the linguistic and reasoning capabilities of LLMs not only for conversational tasks but also for complex planning and decision-making processes (Liu et al., 2023; Singh et al., 2023; Wang et al., 2023c).

Planning, in the context of LLM agents, refers to the process of devising a sequence of actions required to achieve a specific goal. This process inherently relies on the reasoning and decision-making capabilities of LLMs, which are rooted in their ability to understand, generate, and manipulate natural language. For instance, achieving the goal of brushing one's teeth involves a series of steps such as heading to the sink, locating toothpaste, picking up the toothbrush, etc. If a subtask, such as locating toothpaste, fails, the agent must adapt by either setting a new goal (e.g., purchasing toothpaste) or skipping ahead to the next executable step. While LLMs have demonstrated success in planning tasks, challenges remain, particularly in scenarios involving long-horizon goals or complex sequences of actions. As the number of required actions increases, the accuracy of LLM-based planning tends to decline significantly (Valmeekam et al., 2024b). This is because long-horizon tasks expand the search space, and approximate retrieval-based reasoning—typical of current LLMs—struggles to maintain coherence and robustness over extended sequences.

This issue highlights the need for a framework that enhances the robustness of LLMs in solving long-horizon tasks within planning scenarios, while also improving their efficiency in utilizing current conditions to create effective plans. To tackle this challenge, we draw inspiration from human cognition, particularly the ability to break down complex problems into simpler, manageable subproblems. Cognitive psychology, such as that by Simon & Newell (1971); Chipman et al. (2000) suggests that humans naturally decompose difficult tasks into smaller, sequential steps, facilitating reasoning and execution . By mimicking this strategy, LLMs can construct hierarchical plans, enabling more robust and efficient solutions to complex goals.

In this study, we introduce a planning framework that leverages human-inspired decomposition to enhance LLMs'planning capabilities. While most prior methods rely on forward reasoning, our approach is based on backward reasoning, which decomposes goals into subtasks in a top-down manner. As shown in Figure 1, the framework incorporates Backward Reasoning, a strategy well-

(a) Existing method using Forward Reasoning     (b) DRIP using Backward Reasoning

Figure 1: Overview Diagram of the DRIP Concept (Right). The left side illustrates the structure of existing methods using forward reasoning, while the right side represents the proposed method utilizing backward reasoning.

suited to the hierarchical nature of goal decomposition, to achieve both efficiency and robustness in planning.

The contributions of this paper are as follows:

- We propose a planning framework that mimics human-like hierarchical goal decomposition, leveraging LLMs'natural language reasoning for task breakdown.
- Improved robustness in planning tasks: Through experiments on BlockWorld and Minecraft, we demonstrate that the proposed framework enhances the robustness of LLM agents compared to existing methods.

## 2 RELATED WORK

### 2.1 LLM REASONING WITH DECOMPOSE

The ability to simplify complex tasks by breaking them down into smaller, manageable subtasks is a hallmark of human cognition (Chipman et al., 2000). This concept, deeply rooted in cognitive psychology (Simon & Newell, 1971) and logic, has inspired recent advancements in multi-step reasoning using LLMs (Xue et al., 2024; Junbing et al., 2023; Zhou et al., 2023). These studies commonly employ decomposition strategies, where a complex question is divided into simpler sub-questions, solved iteratively, and integrated to achieve the final solution. This approach often aligns with backward reasoning, a process of reasoning from the goal state to the initial state.

Empirical results from these studies have demonstrated significant improvements in the accuracy of solving challenging reasoning tasks. For instance, Xue et al. (2024) reported not only enhanced accuracy but also increased efficiency in reasoning tasks through decomposition. These findings suggest that decomposition-based reasoning is a promising approach for addressing the limitations of LLMs in handling complex problems. Building on this foundation, our study extends the application of backward reasoning from question-answering tasks to planning tasks.

### 2.2 REGRESSION PLANNING

Backward reasoning, or regression planning, has long been studied in classical AI planning literature. It has played a central role in traditional planning algorithms, dating back to early works such as Waldinger (1977). Regression planning involves reasoning backward from the goal state to identify the sequence of actions required to achieve it. However, traditional regression planning methods often rely on symbolic planners, which necessitate predefined causal relationships between actions (Xu et al., 2019; Silver et al., 2022). This reliance on symbolic representations poses significant challenges for real-world applications, where the dynamics of the environment are often too complex or uncertain to be fully captured by static, predefined rules.

In contrast, LLMs offer a unique advantage in their ability to dynamically generate and adapt rules based on their extensive pre-trained knowledge. This generative capability enables LLMs to over-

come the rigidity of symbolic approaches, making them more suitable for real-time applications. Our study leverages this strength of LLMs to implement a regression planning framework that dynamically decomposes goals into sub-goals, addressing the limitations of traditional symbolic methods.

## 2.3 PLANNING FOR LLM AGENTS

Planning methods for LLM agents have been extensively studied, with various approaches proposed to enhance their reasoning and decision-making capabilities. According to the taxonomy by Huang et al. (2024), our study falls under the category of task decomposition, a strategy that has been widely adopted in LLM-based planning.

Forward reasoning approaches, including Chain-of-Thought (CoT) prompting(Wei et al., 2023; Kojima et al., 2023), Plan-and-Solve framework (Wang et al., 2023b), and ReAct (Yao et al., 2023), have significantly enhanced planning capabilities by breaking down problems into subtasks. However, forward reasoning faces inherent challenges in handling complex tasks due to the exponential growth of the search space (Yu et al., 2023), with advanced models struggling to achieve robust performance in long-horizon planning tasks (Valmeekam et al., 2024b).

Backward reasoning has recently been explored in the context of LLM agent planning. For example, Ren et al. (2024) proposed "flipping" the initial and goal states to simulate backward reasoning. While promising, this approach encounters limitations in scenarios with multiple goal states or ambiguous goal representations. For instance, in environments like BlockWorld, a goal such as "*The red block is on top of the blue block*" may allow for multiple valid configurations, leading to inconsistencies in the generated plans.

To address these challenges, our study proposes a stricter adherence to backward reasoning by explicitly decomposing the goal into intermediate subtasks. This approach ensures that each subtasks is well-defined and contributes directly to achieving the final objective. By leveraging the extensive knowledge embedded in LLMs, our framework can handle ambiguous or underspecified goal representations, enhancing its applicability to diverse and dynamic problem-solving contexts.

## 3 PLANNING FRAMEWORK:DRIP

Building upon cognitive psychology and logical reasoning, this study introduces DRIP — a framework that integrates hierarchical decomposition with dynamic planning for LLM agents. Inspired by the theory that humans solve problems by breaking them into subtasks (Simon & Newell, 1971; Chipman et al., 2000), DRIP operationalizes this mechanism through structured backward reasoning. This decomposition process aligns closely with the principles of backward reasoning, enabling the systematic breakdown of high-level goals into actionable subtasks. A high-level overview of the algorithm is presented in Algorithm 1, followed by detailed descriptions of each phase in the subsequent subsections.

### 3.1 DECOMPOSE

In DRIP, "decomposition" refers to the process where the LLM recursively breaks down a goal into subtasks by identifying the necessary preconditions. This forms a reasoning tree, where each node $n \in N$ stores a goal or subtask that represents an action or a condition to be achieved. Let the goal be $g_{00}$ and the condition be $S_0$, and the others are defined as follows.:

- $\pi$(Plan): A list of actions leading from $S_0$ to $g_{00}$.
- $\mathcal{F}$: The set of nodes currently being processed, i.e., nodes whose executability remains false. In Figure 2's example, these are the nodes highlighted with the light blue area.
- $\mathcal{F}_{\text{pending}}$: Temporary set of nodes awaiting execution evaluation. In Figure 2, these are the nodes in the light orange area.
- $\mathcal{F}_{\text{remaining}}$: Set of nodes that failed executability check in current iteration. In Figure 2, these are the node in the light green area.
- $g_{dj}$: Subtasksrequired to achieve the parent node($n_{dj}$)'s goal. $d$ represents the depth from the root node, and $j$ depends on the number of subtasks decomposed by the LLM from the same parent node.

---

**Algorithm 1** DRIP Planning Algorithm

---

**Require:** Initial condition $S_0$, Goal $g_{00}$
**Ensure:** Plan $\pi$ : a sequence of executable actions
1: Initialize reasoning tree with root node $t \leftarrow 0, S_t \leftarrow S_0, n_{00} = \{g_{00}, \texttt{false}\}, \pi \leftarrow [\,], \mathcal{F} \leftarrow n_{00}$
2: **for** $d = 1$ **to** $MAX\_DEPTH$ **do**
3:     **for all** $n_{d-1}$ **do**
4:       $\mathcal{F}_{\text{pending}} \leftarrow \emptyset$
5:       $\{g_{d0}, g_{d1}, \ldots, g_{dk}\} \leftarrow \text{decompose}(n_{d-1}), \mathcal{F}_{\text{pending}} \leftarrow \{g_{d0}, g_{d1}, \ldots, g_{dk}\}$
6:       **while** $\mathcal{F}_{\text{pending}} \neq \emptyset$ **do**
7:         $\mathcal{F}_{\text{remaining}} \leftarrow \emptyset$
8:         **for** $j = 0$ **to** $k$ **do**
9:           **if** executability$(g_{dj}, S_t) = \texttt{true}$ **then**
10:            $\pi \leftarrow \pi \cup \{g_{dj}\}, S_{t+1} \leftarrow \text{apply}(g_{dj}, S_t), t \leftarrow t + 1$
11:            Add $\{g_{dj}, \texttt{true}\}$ as child of $n_{d-1}$ in tree $T$
12:           **else**
13:            $\mathcal{F}_{\text{remaining}} \leftarrow \mathcal{F}_{\text{remaining}} \cup \{g_{dj}\}$
14:           **end if**
15:         **end for**
16:         **if** $\mathcal{F}_{\text{remaining}} = \mathcal{F}_{\text{pending}}$ **then**
17:           **break**
18:         **end if**
19:         $\mathcal{F}_{\text{pending}} \leftarrow \mathcal{F}_{\text{remaining}}$
20:         Add all $\{g_{dj}, \texttt{false}\}$ for $g_{dj} \in \mathcal{F}_{\text{remaining}}$ as children of $n_{d-1}$ in tree $T$
21:       **end while**
22:       **if** $\forall g_{dj} \in \{g_{d0}, g_{d1}, \ldots, g_{dk}\} : \text{executability}(g_{dj}, S_t) = \texttt{true}$ **then**
         $\mathcal{F} \leftarrow \text{checkParentExec}(n_0)$
23:       **end if**
24:     **end for**
25:     **if** executability$(g_{00}, S_t) = \texttt{true}$ **then**
26:       **break**
27:     **end if**
28:     $\mathcal{F} \leftarrow \mathcal{F} \cup \mathcal{F}_{\text{remaining}}$
29: **end for**
30: **return** $\pi$

---

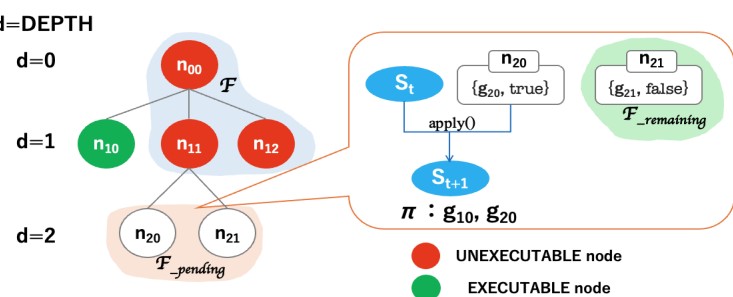

Figure 2: Variable relationships in the algorithm.A detailed symbol table is provided in the Appendix A.1.

At each step, the LLM is prompted to generate subtasks for a given parent node:
$$\{g_{d1}, g_{d2}, \ldots, g_{dk}\} \leftarrow \text{decompose}(n_{d-1})$$
For readability, we write decompose$(n_d)$ as shorthand for decompose_g(subtask$(n_d)$). In other words, when we write decompose$(n_d)$, it means that we decompose the subtasks contained in nodes at depth level $d$. For example (Figure 3), consider the following initial condition from the Block-World dataset(Valmeekam et al., 2023a):

$S_0$: *"The yellow block and orange block are clear, the hand is empty. The orange block is on the table, the blue block is on top of the red block, and the yellow block is on top of the blue block."*(Initial condition in Figure 3 (upper right))

$g_{00}$: *"The red block is on top of the orange block and the yellow block is on top of the red block."*(Goal in Figure 3 (upper left))

As shown in the light gray box of Figure 3, this $g_{00}$ can be decomposed into the actions "Stack red orange" and "Stack yellow red".

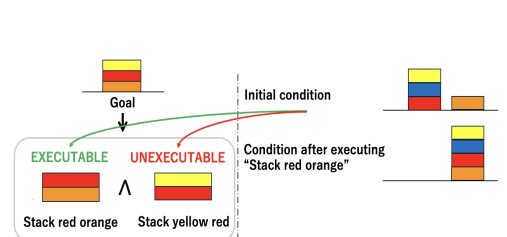

Figure 3: Evaluate subtask executability from the condition and execute the executable ones.

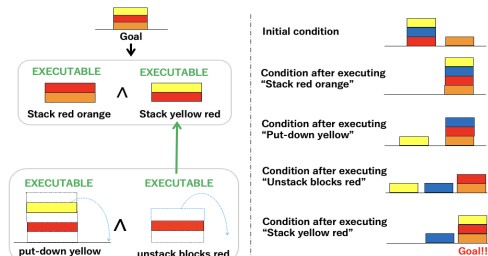

Figure 4: Execute parent nodes when all children are executable. Repeat until reaching the root to complete planning.

## 3.2 EXECUTABILITY

The executability step evaluates whether each subtask can be performed given the current condition. We define the function:

$$\text{executability}(g_{dj}, S_t) \in \{\texttt{true}, \texttt{false}\}$$

The apply function takes a subtask $g_{dj}$ and a condition $S_t$ as input and returns a new condition $S_{t+1}$. In other words, it represents the execution of a valid action by the actuator.

$$S_{t+1} \leftarrow \text{apply}(g_{dj}, S_t)$$

Consider the example from Figure 3: initially, "Stack red orange" is executable, but "Stack yellow red" is not because the red block is not clear. Therefore, as shown in Figure 3, the executability of the subtasks in the initial condition is labeled as EXECUTABLE and UNEXECUTABLE, respectively. Upon executing the former, the condition updates, triggering a reevaluation of pending subtasks. This process is repeated until no executability changes remain.

## 3.3 RE-DECOMPOSITION AND TERMINATION OF TREE CONSTRUCTION

When executability updates stall, any remaining unexecutable actions are reinterpreted as subtasks and recursively decomposed. For example, to execute "Stack yellow red", the LLM infers prerequisite subtasks like "Put-down yellow" and "Unstack blocks red"(the bottom light gray box in Figure 4).

A node's decomposition is complete when all its child subtasks become executable. Once this occurs, executability propagates upward—if all children of a parent node are executable, the parent becomes executable as well. This process programmatically searches for nodes with all EXECUTABLE child nodes, traversing from the current depth toward the root node. Upon finding such nodes, it automatically applies the apply function to update the condition and the node's executability. This operation gradually reduces $\mathcal{F}$, ultimately completing the decomposition process. In the algorithm, this workflow is defined as checkParentExec(). As shown Figure 4, this process continues until the root node is executable, indicating that the original goal can now be achieved.

# 4 EXPERIMENT

## 4.1 BLOCKWORLD

The BlockWorld task involves stacking blocks to achieve a specified goal state, making it a widely studied problem in classical planning. For this study, we utilized the BlockWorld_hard dataset (Valmeekam et al., 2023b; 2024a), which includes scenarios with stacking tasks involving between 6 and 15 blocks. This dataset is particularly challenging due to the increased complexity of the goal states and the number of actions required to achieve them. Detailed statistics regarding the number of blocks and configurations in the dataset are provided in Appendix A.2.1.

### 4.1.1 EXPERIMENT SETUP

We adapted the experimental setting to create a more challenging planning scenario. While maintaining the core BlockWorld dynamics, we made two key modifications: (1) consolidated the action

Table 1: Performance comparison on BlockWorld dataset. P-values computed using Fisher's exact test against DRIP.

| Methods | Claude 3.7 Sonnet[1] | | | | GPT-4o(OpenAI et al., 2024) | | | |
|---------|----------|--------|---------|------------|----------|--------|---------|------------|
| | Accuracy | 95% CI | p-value | Odds Ratio | Accuracy | 95% CI | p-value | Odds Ratio |
| DRIP | **40.9%** (**45**/110) | 31.7-50.1% | - | - | 16.4% (18/110) | 9.5-23.3% | - | - |
| CoT (Kojima et al., 2023) | 23.6% (26/110) | 15.7-31.6% | p < 0.01 | 2.24 | 13.6% (15/110) | 7.2-20.0% | p > 0.05 | 1.24 |
| ReAct (Yao et al., 2023) | 9.1% (10/110) | 3.7-14.5% | p < 0.001 | 6.92 | 1.8% (2/110) | 0.0-4.3% | p < 0.001 | 37.4 |

P-values from Fisher's exact test comparing each method to DRIP within the same model. Numbers in parentheses indicate (successful instances / total instances).

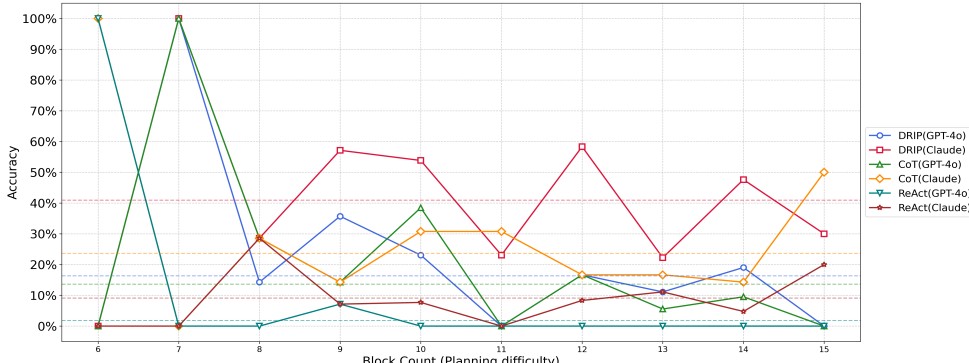

Figure 5: Experimental results. The horizontal axis represents the number of blocks, while the vertical axis indicates the accuracy for each block count. The blue is DRIP (GPT-4o), the red is DRIP (Claude), the green is CoT (GPT-4o), the yellow is CoT (Claude), the cyan is ReAct (GPT-4o), and the brown is ReAct (Claude). The dotted lines indicate the overall accuracy for each method.

space to three essential operations: "Stack [blockA] [blockB]," "Put-down [block]," and "Unstack blocks [block]," and (2) modified the single-block holding constraint to allow simultaneous manipulation of multiple blocks. The latter modification substantially expands the search space by increasing the number of valid actions available at each planning step, thereby creating a more demanding test of planning robustness compared to the standard constrained setting. The experimental settings, including the prompts used for the LLM, are fully described in Appendix A.2.3. All experiments were conducted in Japanese.

### 4.1.2 BENCHMARK

We evaluated DRIP against baseline methods summarized in Table 1. For comparison, we include CoT (Kojima et al., 2023) and ReAct (Yao et al., 2023), which alternates between reasoning and acting. These methods represent fundamental approaches to forward reasoning-based planning, making them ideal baselines for evaluating our backward reasoning framework. By comparing against these established forward reasoning techniques, we aim to investigate whether the robustness advantages of backward reasoning observed in prior research in reasoning extend to LLM-based planning tasks, particularly in complex scenarios where forward search may encounter exponential branching challenges. We employed GPT-4o and Claude 3.7 Sonnet[2] as our models.

Additionally, we include Manual versions where humans execute the proposed actions while LLMs handle only the planning component. These conditions isolate pure planning capabilities from execution errors, allowing us to evaluate the theoretical upper bounds of each approach and assess whether performance differences stem from planning quality or implementation limitations. Detailed methodology for the manual execution experiments is provided in the Appendix A.2.2.

### 4.1.3 RESULTS

The experimental results are summarized in Table 1 and Figure 5. DRIP demonstrates statistically significant improvements over baseline methods, with performance varying substantially across dif-

---

[2]https://www.anthropic.com/claude/Sonnet

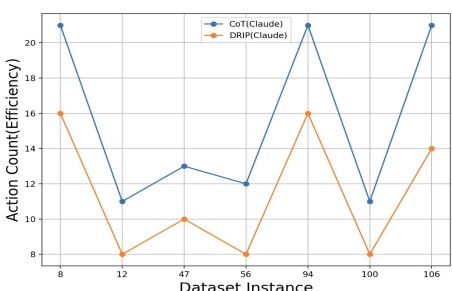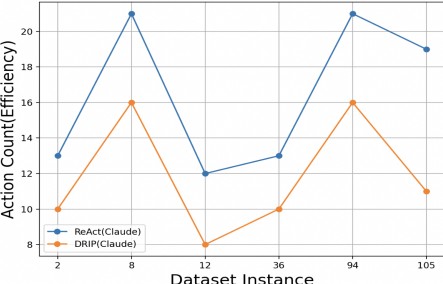

Figure 6: The difference in the number of actions included in the planning. The left side shows CoT vs DRIP, and the right side shows ReAct vs DRIP. In both cases, orange represents DRIP and blue represents the comparison method. The horizontal axis represents the instance numbers correctly solved by both methods, while the vertical axis represents the number of actions included.A lower action count indicates more efficient planning.

ferent language models. Using Claude 3.7 Sonnet, DRIP achieved 40.9% accuracy, significantly outperforming CoT (23.6%, $p < 0.01$, odds ratio = 2.24) and ReAct (9.1%, $p < 0.001$, odds ratio = 6.92). With GPT-4o, DRIP maintained its advantage over ReAct (16.4% vs 1.8%, $p < 0.001$, odds ratio = 37.4), though the difference with CoT was not statistically significant (16.4% vs 13.6%, $p > 0.05$).

To assess planning efficiency, we analyzed the number of actions used by DRIP (Claude) compared to baseline methods in successful cases (Figure 6). DRIP consistently demonstrated superior efficiency, requiring on average 4.29 fewer steps than CoT (Claude) and 4.67 fewer steps than ReAct (Claude). This efficiency advantage stems from DRIP's backward reasoning approach, which decomposes goals only when necessary and avoids the exhaustive step generation characteristic of forward reasoning methods.

To isolate pure planning capabilities from execution errors, we evaluated manual execution conditions where humans performed the actions proposed by GPT-4o. DRIP (Manual) achieved 82.7% accuracy compared to ReAct(Manual) at 31.8%, confirming that the performance advantages stem from superior planning quality rather than implementation artifacts. The main reason why DRIP (Claude) could not match the performance of DRIP (Manual) lies in the difficulty of accurately describing block conditions using natural language such as "block X is clear" or "block Y is on block Z." The decline highlights a key limitation: as task complexity grows, condition descriptions become verbose and ambiguous, leading LLMs to misjudge action executability. This suggests that while DRIP's planning framework is sound, future improvements may require multimodal inputs or formal representations like PDDL.

### 4.1.4 ERROR ANALYSIS

Table 2: Details of DRIP (Manual) 's error type.

| Error type | Number of datasets |
|---|---|
| Errors in decomposition by LLMs | 14 |
| Errors in the Framework | 5 |
| total | 19 |

We analyzed the 19 cases (17.3% of the total) where DRIP (Manual) failed, summarized in Table 2. Of these, 14 errors stemmed from incorrect decomposition by the LLM. Despite the structured nature of BlockWorld and clear prompts, the model occasionally generated invalid action sequences, especially in configurations with ambiguous or complex block relationships.

The remaining 5 errors were found to be caused by fundamental limitations of the current framework. These errors highlight scenarios where the framework's reliance on goal-based reasoning alone is insufficient. For example, consider a goal is: "*block a is on top of block j, block b is on top of block d, block c is on top of block b, block d is on top of block a, block f is on top of block i, block g is on top of block f, block i is on top of block c, block j is on top of block h*" (i.e., 'g-f-i-c-b-d-a-j-h'). Suppose the current condition is: 'c-b-d-a-j-h-e-g-f-i'. In this case, the remaining action to achieve the goal is "stack i c". Decomposing this action requires clearing block 'c' and moving it to create a

separate tower with 'i' and 'c'. However, creating such a separate tower is not feasible because the goal condition ('c-d-b-a-j-h') has already been partially achieved. Moving block 'c' would violate the goal condition, making it impossible to proceed without undoing previously achieved subtasks. This example illustrates a key limitation of the current framework: it considers actions solely based on the goal state and does not account for the constraints imposed by the current condition. In certain scenarios, achieving the goal requires reasoning that integrates both the goal state and the current condition, as well as the ability to dynamically adjust the plan to avoid conflicts between intermediate subtasks.

## 4.2 MINECRAFT

In the previous section, it was demonstrated that our framework can be applied to planning by utilizing classical planning problems. However, real-world tasks are far more dynamic and underspecified. To explore DRIP's applicability in such settings, we conducted experiments in Minecraft[3]. Minecraft is an open-world 3D sandbox game that enables flexible and complex tasks similar to those in the real world, such as diamond mining and farming. Agents can be controlled through JavaScript code[4], and it has been utilized in numerous studies as a platform for evaluating agent performance (Fan et al., 2022; Wang et al., 2023a; Zhao et al., 2024; Wang et al., 2023d).

### 4.2.1 EXPERIMENT SETUP

We conducted experiments using DRIP, ReAct, and CoT as planners. We used Claude 3.5 Sonnet from the Claude series, which showed good results in BlockWorld The task is specified as "difficult" in the Minedojo dataset, which involves mining diamonds from barehand conditions. ReAct repeatedly provides the goal "mine diamond" while having the LLM output the next subtask based on successfully completed subtasks and current observation information, then the agent acts on that subtask. After completing an action, it repeats the loop of considering the next subtask again A maximum of 70 iterations is performed, and if diamond can be mined during this period, the task is considered successful. For CoT, we conducted experiments using a Plan-and-Solve(Wang et al., 2023b) approach where the system is made to think of a plan necessary for the "mine diamond" task and then execute that plan sequentially.

For agent action generation, we used Claude Sonnet 3.5 to generate JavaScript code. Following Voyager(Wang et al., 2023a)'s approach, we utilized code from Voyager's Skill Library directory to perform in-context learning for generation. The process flow is as follows: after each planner decides on the next action, an LLM selects the most applicable code from the Skill Library for that action. Subsequently, the LLM generates action code by incorporating the selected code into the prompt. This code is then executed in the environment. After execution, the LLM determines whether the operation failed based on environmental information. In case of failure, error codes and other outputs are included in the prompt, and the LLM reconsiders the code. If this process fails even after a maximum of 5 iterations, the action proposed by the planner is considered to have failed. Experimental details including prompts are also provided in Appendix A.3.

### 4.2.2 RESULTS

The success rates for each resource are summarized in Table 3. DRIP consistently achieved the highest success rates across all resource types, maintaining 100% completion for wood, stone, and iron tasks, and achieving 80% even for the most challenging mining diamond task. This demonstrates DRIP's robustness in long-horizon tasks. Although ReAct achieved high success rates, it succeeded in diamond mining only once out of five experimental trials. CoT completely failed in mining diamond (0%).

We analyzed the average number of subtasks required for each planning method to complete each task (Table 4). This analysis revealed that, unlike in BlockWorld, DRIP was not necessarily the most efficient method in Minecraft. For subtasks up to stone pickaxe, which all methods achieved 100% completion, CoT showed the smallest number of subtasks despite having the lowest success rate for diamond mining. In diamond mining, DRIP's average number of subtasks upon success (37.25) proposed more subtasks compared to ReAct's only successful case (27). This is because

---

[3]https://www.minecraft.net

[4]https://github.com/PrismarineJS/mineflayer

Table 3: Success rate for each resources.

| Planner | Wood | Stone | Iron | Diamond |
|---------|------|-------|------|---------|
| DRIP | 100%(5/5) | 100% (5/5) | 100%(5/5) | 80%(4/5) |
| ReAct | 100%(5/5) | 100%(5/5) | 80%(4/5) | 20%(1/5) |
| CoT | 100%(5/5) | 100%(5/5) | 60%(3/5) | 0%(0/5) |

Table 4: Average number of steps to reach each resource. For ReAct's diamond task only, this represents the count from a single successful attempt.

| Planner | Wood | Wood Pickaxe | Stone | Stone Pickaxe | Iron | Iron Pickaxe | Diamond |
|---------|------|--------------|-------|---------------|------|--------------|---------|
| DRIP | 1.0 | 5.4 | 8.6 | 11.2 | 15.4 | 34.0 | 37.25 |
| ReAct | 1.0 | 4.0 | 5.0 | 6.4 | 16.5 | 23.3 | 27.0 |
| CoT | 1.0 | 4.2 | 5.2 | 7.4 | 11.0 | - | - |

ReAct and CoT planning propose forward-looking subtasks, such as logging enough wood that will likely be needed later, taking the overall task into account. Since DRIP only proposes the number required for the immediate parent node, it may repeatedly perform basic tasks like "mining wood," resulting in a higher average number of subtasks. However, because the subtasks proposed by ReAct and CoT are forward-looking and comprehensive, each individual subtask becomes a longer-horizon task compared to DRIP, leading to more frequent execution failures and lower success rates. DRIP's fine-grained subtask proposals allow for steady progress in action generation, which is considered to contribute to its higher success rate in diamond mining. Each experimental result of DRIP is discussed in more detail in Appendix A.3.2.

## LIMITATIONS

The proposed DRIP framework demonstrates robustness and efficiency in planning by mimicking human capabilities. However, it has several limitations. First, there are challenges related to the decomposition capabilities of LLMs. While LLMs possess vast amounts of knowledge, the extent to which they can perform commonsense reasoning remains largely unexplored. For instance, executing an action like "*move A to the position of B*" requires the preconditionthat "*A is located somewhere other than B.*" In this study, we explicitly specified feasible actions and utilized structured tasks in the experiments. However, in real-world applications, this limitation could have a significant impact.

Second, the number of LLM calls required is an issue. While CoT requires a single call, DRIP (Manual) uses hierarchical reasoning, averaging 5.98 calls, and DRIP (Claude) averages 6.18 calls. On the other hand, the average number of LLM calls for ReAct (Manual) is 28.3, whereas DRIP achieves a significant reduction in comparison. Humans are said to switch between different types of reasoning, as exemplified by the "Fast and Slow" theory(Kahneman, 2011). Building on these insights, further exploration is needed to develop methods that appropriately combine backward reasoning and forward reasoning.

## CONCLUSION

This paper proposed a planning framework for LLM agents inspired by human problem-solving, particularly the ability to decompose complex problems into simpler components. By employing a backward reasoning approach, the framework dynamically decomposes tasks into prerequisite subtasks, enhancing planning robustness and aligning with human cognitive processes.

From the experimental results, we found that our framework demonstrates superior robustness compared to forward reasoning-based methods. Particularly in structural tasks such as BlockWorld, it was shown to perform efficient planning with fewer steps by avoiding unnecessary actions. In tasks with high degrees of freedom in actions, such as Minecraft, our method did not always achieve planning with fewer steps. However, through comparison with existing methods, we confirmed a trade-off where reducing the number of steps leads to decreased success rates. In the future, toward extending to real-world applications, we will develop our proposed method to enable efficient planning while maintaining its robustness.

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

# A APPENDIX

## A.1 ALGORITHM

The symbols appearing in the algorithm are summarized in the table below.

Table 5: Algorithm Symbol Notation

| Symbol | Description |
|---|---|
| **Variables** | |
| $S_0$ | Initial condition of the environment |
| $S_t$ | Current condition at time step $t$. |
| $g_{00}$ | Initial goal to be achieved |
| $\pi$ | Plan: sequence of executable actions |
| $t$ | Number of executable actions. |
| $T$ | Reasoning tree structure |
| $n_{dj}$ | Node at depth $d$, index j with structure |
| $n_{00}$ | Root node of the reasoning tree |
| $d$ | Depth level from root node |
| $j$ | Index of subtasks decomposed from same parent (j = 0, 1, …, k) |
| $g_{dj}$ | subtask at depth $d$, index $j$ j |
| $\{g_{d0}, g_{d1}, \ldots, g_{dk}\}$ | Set of subtasks decomposed from parent node |
| $\mathcal{F}$ | Frontier: set of nodes currently being processed |
| $\mathcal{F}_{pending}$ | Temporary set of nodes awaiting execution evaluation |
| $\mathcal{F}_{remaining}$ | Set of nodes that failed executability check |
| **Functions** | |
| decompose($g_{dj}$) | Decomposes subtask $g_d j$ in node $n_d j$ into subtasks |
| executability($g_{dj}, S_t$) | Evaluates if subtask $g$ is executable in condition $S_t$ |
| apply($g_{dj}, S_t$) | Execute subtask $g_{tj}$ at condition $S_t$, returns new condition $S_{t+1}$ |
| checkParentExec($n$) | Checks parent executability when all children are executable |
| $MAX\_DEPTH$ | Maximum allowed depth for tree expansion |
| true | Boolean true value for executability |
| false | Boolean false value for executability |

## A.2 EXPERIMENT(BLOCKWORLD)

### A.2.1 DETAILS OF THE BLOCKWORLD DATASET

We summarize the number of blocks and the number of instances for each dataset in the BlockWorld dataset in a table6.

Table 6: Details of the number of blocks in the dataset

| Number of blocks | Number of datasets |
|---|---|
| 6 | 1 |
| 7 | 1 |
| 8 | 7 |
| 9 | 14 |
| 10 | 13 |
| 11 | 13 |
| 12 | 12 |
| 13 | 18 |
| 14 | 21 |
| 15 | 10 |
| total | 110 |

### A.2.2 DETAILS OF THE BENCHMARK

DRIP (Manual) refers to a method where a human acts as the actuator to stack the blocks. In this method, humans determine executability and provide feedback on whether the actions proposed by the LLM were successfully executed or not. The LLM responsible for action decomposition only reasons about actions based on the goal and humans do not receive feedback on the condition. The termination condition is when the root node action (data set goal) is determined to be executable.

In ReAct (Manual), humans execute the actions proposed by the LLM and return the resulting new state as an observation after each action. In this approach, the goal and initial condition are provided at the beginning, and the LLM generates actions based on this information. After executing an action, humans provide the updated condition to the LLM, which then generates the next action based on the new condition. This cycle continues iteratively.

### A.2.3 PROMPT

The prompts used in the BlockWorld experiments are attached.

Table 7 is the prompt used for decomposition and is utilized in both DRIP (Manual) and DRIP (LLM).

Table 8 and Table 9 are prompts used in DRIP (Claude, GPT-4o) to utilize LLM as an actuator. Table 8 is a prompt used to determine whether an action is executable, while table 9 is a prompt used to describe how the condition of the blocks changes after an action is performed.

I am playing with a block set where I need to stack and organize the blocks into a stack. My ultimate goal is to reach the goal state as efficiently as possible without making mistakes.

My Goal"{task}". Please provide actions based solely on the conditions required to achieve this goal.

Available Actions:
1.stack [block1] [block2]
Place [block1] on top of [block2].
2.unstack blocks [block2]
Remove blocks from the top of [block2] and clear [block2] by placing them on the table.
3.put-down [block2]
Place [block2] on the table.

Constraints and Notes:
- Only when the goal state is in the form of "stack [block1] [block2]", return the following actions:
unstack blocks [block2]
put-down [block2]
- For all other conditions:
Return the action stack [block1] [block2].
- Eliminate unnecessary information:
Always respond with actions in a list format. Do not include extra sentences, structures (e.g., [ ] or parentheses), or numbers. Always adhere to the following format:
action1
action2
action3

[Example Responses]

[Goal]
stack blue yellow

[Actions Based on Conditions]
unstack blocks yellow
put-down yellow
[Goal]
task

[Actions Based on Conditions]

Table 7: Decomposition Prompt for BlockWorld.This is an English translation of the Japanese prompts.

You are playing with a block set where you need to stack and organize the blocks into a stack. Based on the given state, determine whether the specified action is executable.

##BlockWorld Action Rules unstack blocks [block2] - blocks refers to the blocks above [block2]. This action involves removing the blocks above [block2] to clear [block2] and placing them on the table. If you move the blocks above [block2], all blocks stacked on top of them will also be moved together.
put-down [block2] - This action involves placing [block2] on the table to separate [block1] and [block2] into different towers. **All blocks stacked on top of [block2] will also be moved together.**
stack [block1] [block2]
- This action involves stacking [block1] on top of [block2]. [block2] must be clear for this action to be performed. **All blocks stacked on top of [block1] will also be moved together**.

##BlockWorld Rules
- You can hold any number of blocks at once.
- Even if there are blocks above a block, you can pick up the block along with all the blocks stacked on top of it.
- When moving a block, all blocks stacked on top of it must be moved together.
- You cannot stack blocks that are already part of the same tower.M
##Decision Procedure
Analyze the situation using the following steps and explicitly output the results for each step:
1. Analyze the Current State
- List the positional relationships of all blocks in bullet points.
- Check if there are any blocks above each block.
- Check if each block is on the table.
2. Analyze the Desired Action
- Moving Block: Identify which block is being moved.
* The "moving block" refers to the specified base block. However, if there are other blocks stacked on top of this block, all of them must be moved together.
- Destination: Specify whether the destination is a block or the table.
- State of the Destination: If the destination is a block: Check if there are any blocks on top of [block2]. If the destination is the table: It is always possible to place blocks on the table.
3. Determine if the Action is Unnecessary
"Unnecessary Action": The action is unnecessary if any of the following conditions are met:
Action: unstack blocks [block2]
The action is unnecessary if there are no blocks above [block2].
Action: stack [block1] [block2]
The action is unnecessary if [block1] is already stacked on [block2].
Action: put-down [block2]
The action is unnecessary if [block1] and [block2] are already part of different towers.
Whether [block2] is already on the table is irrelevant.
4. Determine Executability (Skip this step if the action is unnecessary)
"Not Executable": The action is not executable if any of the following conditions are met:
If the destination is a block ([block2]), and there are other blocks on top of [block2].
If the moving block ([block1]) and the destination block ([block2]) are part of the same tower.
"Executable": You can always move a block, even if there are blocks stacked on top of it.
You can stack a block on the destination ([block2]) if [block2] is clear.
You can always place blocks on the table.
If the moving block is part of a different tower from the destination block, stacking is possible.
"END":If the action is END.
##Current Target for Evaluation
State: initial_condition
Action: action_list
1.State Analysis:[Describe the positional relationships of all blocks in bullet points here.]
2.Action Analysis:[Describe the state of the source and destination here.]
3.Determine if the Action is Unnecessary
4.Decision Result:[Decision],[Reason] (Separate the decision and reason with a comma)

Table 8: Executability prompt for BlockWorld(Translation to English)

You are playing with a block set where you stack blocks. Consider how the situation will change if you take the action {action} from the situation {condition}, and describe it in writing.
## Block World Action Rules unstack [block1] [block2]
- [block1] refers to the block above [block2]. This action removes [block1] from [block2] and places it on the table, clearing [block2]. If [block1] is moved, all blocks above [block1] move with it. [block2] is cleared.

put-down [block2]
- This action places [block2] on the table to separate [block1] and [block2] into different towers. **All blocks on top of [block2] move together.** [block2] is cleared and placed on the table.
stack [block1] [block2]
- This action places [block1] on top of [block2]. [block2] must be cleared. **All blocks above [block1] will move together.**
## Block World Rules
- You can have as many blocks as you want.
- **Blocks can be picked up together from the bottom even if there are other blocks on top of them.**
- When moving a block, **all blocks above it will move together.**
- You cannot stack blocks that are in the same tower.
- If the destination block ([block2]) is not clear, you cannot stack [block1] [block2].
- An empty/clear block means that there is nothing on top of it.

As your final answer, describe the empty blocks, the state of the blocks, and the blocks on the table in the same order as the given situation. Use the format "X is in Y" consistently.
Answer in the following format.
## Current target for judgment
State: {condition}
Action: {action}
Thought process:
Final state: [answer ]

Table 9: The prompt for updating conditions for BlockWorld. This is an English translation of the Japanese prompts.

## A.3 EXPERIMENT(MINECRAFT)

### A.3.1 PROMPT & AGENT'ENVIRONMENT INFORMATION

The prompts used in the Minecraft experiment are attached below. The task decomposition prompt is Table 10. In the Minecraft experiment, there are 8 types of subtasks that the Agent can perform: "craft", "mine", "smelt", "kill", "cook", "equip", "explore", and "place". These were selected from actions defined as primitives in Voyager.

The prompt for determining whether a task can be executed is Table 11. In the Minecraft world, tasks that were already achieved during the execution of other nodes were frequently observed, so we also determined "Achieved" status. This includes subtasks that are already completed and therefore require neither code generation nor decomposition.

The prompt for code generation is Table 12. This prompt is a modified version of Voyager's action_template.txt[5]. For {sample_code}, we had the LLM select the most useful code from Voyager's skill_library[6], attached that code, and used it for few-shot learning. The bot's environmental information used for executability assessment([Current bot status]:{new_observation}) and code generation([Environment Settings]:{observation}) includes the following information.

Basic Information

- Position: Bot's current coordinates (X, Y, Z)
- Biome: Current location's biome name
- Time Period: Day/night determination
- Health: Bot's current HP

---
[5]https://github.com/MineDojo/Voyager/blob/main/voyager/prompts/action_template.txt
[6]https://github.com/MineDojo/Voyager/tree/main/skill_library/trial1/skill/code

- Inventory: List of possessed items (quantity and name)
- Main Hand Item: Currently held item in main hand

Resource Information (Within 15 blocks)

- Wood: Distance to nearest log and total count
- Stone: Distance to nearest stone block and total count
- Ore: Distance to nearest ore and total count
- Crafting Table: Availability status
- Furnace: Availability status

Nearby Placed Blocks (Within 5 blocks) Detailed information with positions for:

- Crafting Tables: Position, distance, count
- Furnaces: Position, distance, count
- Chests: Position, distance, count
- Workstations: Anvils, enchanting tables, brewing stands, etc.

Task-Specific Information Determined from inventory items:

- Material Possession: Availability of wood and stone materials
- Tool Possession: Detailed information on pickaxes, axes, swords, shovels, hoes
- Equipment Items: Possession of crafting tables and furnaces

### A.3.2 DETAIL OF RESULTS

This appendix presents detailed results from DRIP's Minecraft experiments. Figure 7 shows the data from five DRIP experiments as a mean ± standard deviation graph. DRIP reached the diamond mining task in a minimum of 21 steps and a maximum of 57 steps. In the one failed attempt to mine diamonds, the agent died twice before mining stone and before smelting iron to create an iron pickaxe, resulting in an empty inventory. This caused the acquisition of stone and subsequent resources to require significantly more steps. However, this observation demonstrates that even when a task fails midway, DRIP can resume planning without losing sight of the goal. In the future, we plan to conduct experiments with more realistic tasks to explore a better balance between efficiency and robustness.

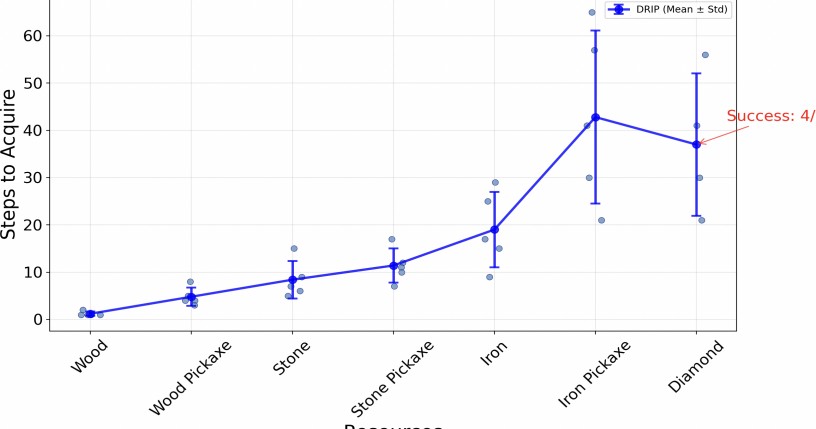

Figure 7: Steps required to acquire each resource in Minecraft using DRIP (n=5 experiments). Error bars show standard deviation. Individual points represent raw experimental data.

## Role:
You are a helpful assistant that suggests actions to take to accomplish a task in Minecraft.

## Instructions:
Break down the requirements to achieve the following goal "{task}" into only the immediately preceding step for this task.
## Constraints:
1: Only list what is necessary for **the immediately preceding step to efficiently achieve the goal** of this task.
2: **Do not include earlier steps.** Example: "Craft an iron pickaxe" -> "Craft 3 iron ingots, craft 2 sticks, place a crafting table" (do not include mining iron ore).
3. Suggest a slightly extra number of items.
4. The available actions are: Table .
5. The "place" action requires the "craft" action as a prerequisite. 6. "Mining" or "killing" something requires the "explore" action to find something (e.g., explore to the location of wood blocks, explore to the location of iron ore blocks).
7. If items are needed, write the specific item name. Example: Wooden Pickaxe, Iron Armor (Pickaxe, Armor alone are insufficient).
8. The executing bot does not require light or food.
9. Only provide concrete actions. Do not include supplementary information.
10. Respond only in a list format.

## Order of response rules:
- Gather necessary materials (mine, kill, craft)
- Prepare equipment (place, equip)
- Execute (explore, smelt, cook)

## Response format:
[Goal]
{task}

[Immediately preceding step ]

Table 10: Prompt for subtask decomposition in Minecraft. Translated into English.

## Role:
You are an expert agent determining the executability of tasks in the Minecraft world. Consider this based on your knowledge of Minecraft.

## Instructions:
Determine whether the specified task is executable in the current situation or if it has already been achieved.

## Criteria:
1. Achieved (Done):
- The task is fulfilled (e.g., if the task is "equip [item]," and [item] is in the main hand).
- The item resulting from the task is already in the inventory/equipment, and the required quantity is met.
- The task's destination has already been reached.

2.Executable (True): The task can be directly executed at the current location.
- Necessary blocks/entities are nearby.
- It can be inferred as executable from the Biome or surrounding conditions.
- **Underground exploration tasks are feasible even from above ground if appropriate tools (e.g., pick-axe) are available.**
- **Wood blocks, dirt, sand, gravel, and leaves can be mined with bare hands.**
- **Specific quantities will be confirmed at the time of execution.**
- **The task "explore to the location of [item]" is feasible if basic equipment to find that item is available.**

3.Unexecutable (False): Necessary conditions are missing.

## Important:
- Whether a task is achieved or not should be determined by the resulting item/situation, not by the verb of the task (e.g., chop, kill, mine).
- Do not consider efficiency. Only determine feasibility.

## Response Format:
- If achieved: "Done: [Reason]"
- If executable: "True: [Reason]"
- If not executable: "False: [Reason]"

[Task]: {task}
[Current bot status]: {new_observation}
[Output]: {stdout}

[Judgement]:

Table 11: Prompt for judging task executability. Translated into English.

1026
1027
1028
1029
1030
1031
1032
1033
1034
1035
1036
1037
1038
1039
1040
1041
1042
1043
1044
1045
1046
1047
1048
1049
1050
1051
1052
1053
1054
1055
1056
1057
1058
1059
1060
1061
1062
1063
1064
1065
1066
1067
1068
1069
1070
1071
1072
1073
1074
1075
1076
1077
1078
1079

## Role
You are an agent that creates Mineflayer JavaScript code to accomplish Minecraft tasks specified by me.
Below is a useful program written using the Mineflayer API, which is highly relevant to the task. Please use it as a reference.
{sample_code}

Please provide the necessary function code that meets the following conditions and follows the proposed action plan:

## Conditions
- The bot has already been created and exists in the Minecraft game world.
- The bot does not require light or food.
- Do not include anything that is already in the code header.
- No need to set up the bot instance.
- Write an asynchronous function that takes the bot as its only argument.
- Avoid using await as much as possible.
- Use 'mineBlock(bot, name, count)' to collect blocks. Do not use 'bot.dig' directly.
- Use 'craftItem(bot, name, count)' to craft items. Do not use 'bot.craft' or 'bot.recipesFor' directly.
- Use 'smeltItem(bot, name, fuelname, count)' to smelt items. Do not use 'bot.openFurnace' directly.
- Use 'placeItem(bot, name, position)' to place blocks. Do not use 'bot.placeBlock' directly.
- Use 'killMob(bot, name, timeout)' to kill mobs. Do not use 'bot.attack' directly.
- Use 'bot.chat' and 'console.log' to report progress in the chat and console.
- If something cannot be found, use 'exploreUntil(bot, direction, maxDistance, callback)'. You must call this frequently before mining blocks or killing mobs. You should choose a random direction each time (do not always use (1, 0, 1)).
- 'maxDistance' must always be 32 for 'bot.findBlocks' and 'bot.findBlock'. Do not cheat.
- Do not write infinite loops or recursive functions.
- Do not use 'bot.on' or 'bot.once' to register event listeners. They are absolutely unnecessary.
- After calling the function, always call 'bot.quit();' to log out the bot.
- At the end, use 'bot.on('spawn', () => );' to call the function.

Write the necessary Mineflayer JavaScript function to complete the specified task "{task}".
Also, include the specific steps and their explanations that will be performed within the function.

[Environment Settings]:
{observation}

[Task]:
{task}

[Action Plan]:
{action}

[Code Header]:
{js_setting}
[Code]:

Table 12: Prompt for Minecraft code generation. Translated into English.