# OpenReview forum: "DRIP: Decompositional reasoning for Robust and Iterative Planning with LLM Agent"
_ICLR.cc/2026/Conference — Submitted to ICLR 2026_

### Official Review · Reviewer_Py5F · 2025-10-25

**Soundness:** 3
**Presentation:** 4
**Contribution:** 2
**Rating:** 4
**Confidence:** 4

**Summary:**

This paper introduces DRIP, a planning framework for LLM agents based on backward reasoning and task decomposition, aimed at enhancing robustness in long-horizon planning tasks. Its core contribution lies in formalizing a human-like problem decomposition mechanism for LLM planning, realized through the construction of a dynamic, goal-driven plan via an executability reasoning tree. Experimental results in both BlockWorld and Minecraft environments demonstrate its superior robustness compared to forward-reasoning baselines.

**Strengths:**

1. **Originality:** The paper presents a systematic implementation of backward reasoning for LLM planning, offering a clear and contrasting alternative to the predominant paradigm of forward reasoning.

2. **Quality:** The proposed method is well-designed and rigorously described. The experimental design is comprehensive, effectively validating the framework across both structured (BlockWorld) and open-world (Minecraft) tasks.

3. **Clarity:** The paper is clearly structured. The inclusion of overview diagrams, detailed algorithm pseudocode, and a comprehensive symbol table greatly aids in understanding the proposed framework.

**Weaknesses:**

1. **Insufficient Experimental Comparison:** The empirical evaluation lacks direct comparisons with other recent backward reasoning methods. This omission makes it difficult to precisely assess the unique advantages and distinctive contributions of DRIP within the landscape of backward reasoning approaches.

2. **Limited Generalizability Validation:** The framework's performance is validated only in the BlockWorld and Minecraft domains. Broader assessment on more diverse and realistic task benchmarks—such as robotic manipulation or everyday planning tasks—is needed to fully establish its general applicability.

3. **Incremental Nature of Contribution:** The core idea of backward reasoning is well-established in classical AI planning. While the work is solid, the primary novelty lies in its effective adaptation and demonstration using LLMs, rather than in introducing a fundamentally new reasoning paradigm.

**Questions:**

* Q1: Have the authors considered a hybrid approach that strategically combines DRIP's backward reasoning with elements of forward reasoning? This could potentially strike a more optimal balance between robustness and planning efficiency, mitigating the observed increase in subtask steps in Minecraft.

* Q2: How does DRIP handle scenarios with ambiguous goal states or multiple concurrent goals? Could the authors discuss the framework's stability in such settings and any potential strategies to address these challenges?

* Q3:Are there plans to evaluate DRIP in more complex simulated environments or on real-world physical tasks? This would significantly strengthen the claims regarding its practicality and robustness for real-world applications.

---

> ### Author Response · Authors · 2025-11-21
>
> Thank you very much for the thoughtful and constructive review. We are glad that Reviewer Py5F found the clarity, motivation, and design of DRIP valuable. Below we address the main concerns.
>
> (1) On hybridizing backward and forward reasoning (Q1)
>
> We appreciate this excellent suggestion. Through our experiments, we also realized that a human-like hybrid strategy—switching flexibly between backward reasoning and forward reasoning—could be beneficial for improving planning efficiency, especially in open-ended environments.
>
> However, determining when to switch between backward and forward reasoning, and how to merge the resulting partial plans, requires substantial design decisions that go beyond the scope of the current study. In this paper, our primary goal is to show that backward reasoning alone, when coupled with executability estimation, already constitutes a robust planning paradigm for LLM agents.
>
> We view hybridization as a very promising direction and intend to explore it as future work.
>
> (2) On ambiguous goal states and multiple concurrent goals (Q2)
>
> In the current implementation of DRIP, both ambiguous goals and multiple concurrent goals are delegated to the LLM during the decomposition stage.
>
> If the reviewer refers to ambiguity stemming from linguistic underspecification or vagueness, we agree this is an important and interesting research problem. Ambiguity resolution in LLM agents often requires pragmatic reasoning, clarification questions, and grounding through human–agent interaction. We consider this tightly connected to research in pragmatics and HRI, and we plan to incorporate such mechanisms in future extensions of DRIP.
> At present, DRIP uses the LLM’s interpretation of the goal as the starting point of decomposition.
>
> In our experiments, we consistently observed that when multiple goals are given (e.g., “Put away toys AND vacuum the floor.”), the LLM naturally outputs an ordered list of subtasks (e.g., “Put away toys, then vacuum the floor.”). DRIP is therefore designed to use this ordering directly. We view this sequencing behavior as an inherent property of LLMs, and the current DRIP framework is intentionally aligned with it.
> Explicit mechanisms for goal disambiguation, prioritization, or multi-goal optimization are promising research topics, and we plan to explore them in future work.
>
> (3) Evaluation in broader domains (Q3)
>
> We agree that expanding evaluation to more diverse environments will further strengthen the generality of DRIP. Our next steps include applying DRIP to robotic manipulation and household-level tasks, where backward decomposition and executability reasoning remain crucial.
>
> (4)On the incremental nature of the contribution(Weakness 3)
>
> We appreciate the reviewer’s observation. As discussed in our related work section, the core idea of backward reasoning has indeed been explored in classical AI planning and cognitive science. Our intention is not to claim backward reasoning itself as a new paradigm, but rather to show that its underlying decomposition principles can be made operational and effective in modern LLM-based planning, which has fundamentally different characteristics from symbolic planners.
>
> In LLM agents, actions are not predefined operators but natural language outputs with uncertain executability. Demonstrating that a classical concept such as backward reasoning can be re-instantiated under these conditions—through dynamic executability estimation and recursive decomposition—constitutes the central contribution of DRIP. In this sense, the novelty lies in adapting and operationalizing a well-established reasoning principle in a setting where it has not been straightforwardly applicable.
>
> We believe this perspective complements the reviewer’s point: DRIP connects classical reasoning ideas with contemporary LLM capabilities, highlighting how established theories can gain new practical relevance in modern agent architectures.
>
>
> We sincerely appreciate the reviewer Py5F’s careful evaluation and insightful suggestions. Many of your comments align directly with our intended long-term research trajectory, and we believe they will substantially enhance future versions of DRIP.
> We hope that the clarifications provided here help convey the value and impact of the present contribution, and we appreciate your consideration in the final decision.

---

### Official Review · Reviewer_aMaG · 2025-10-30

**Soundness:** 1
**Presentation:** 2
**Contribution:** 1
**Rating:** 2
**Confidence:** 5

**Summary:**

The authors consider planning an important problem, but current premier LLMs fall short of generating robust plans. The paper devises a planning process grounded in cognitive psychology, stating that the proposed DRIP framework leverages human-inspired decomposition to enhance LLMs’ planning capabilities. They argue that the novelty and effectiveness of DRIP lie in performing both forward, top-down reasoning and backward reasoning. The introduction is light on technical specifics and on clear statements of novelty; the authors should at least explain why backward reasoning is helpful. The tooth-brushing example is weak and offers little insight.

**Strengths:**

This paper points out that planning is an important problems, and LLMs could be helpful.

It causally reason that backward reasoning can be helpful to reduce computational cost.

Limited studies on "simple" planning problem seems to yield some improvement.  But the paper should step up to deal with serious planning problems such as supply-chain management, etc.

**Weaknesses:**

There are several shortcomings in this paper.

1. Related work coverage. It is not yet comprehensive for a planning paper. The section emphasizes decomposition and regression planning but misses four pillars that serious planners consider essential: uncertainty and belief tracking, plan repair and rollback, tool-grounded interaction, and memory or context management. It also needs a brief evaluation critique.

2. Native LLM limitations unaddressed. Context loss on long horizons is a known problem. Self-validation is inherently limited in light of Gödel’s incompleteness results. The paper does not discuss these issues.

3. Cost of backward search. Backward search can explode when many goal configurations are admissible. What constraints are used (landmarks, goal ordering, HTN templates, causal graphs) to prevent exponential cost? The empirical study should examine efficiency and effectiveness trade-offs.

4. Persistent memory. LeCun has noted that LLMs lack persistent memory and therefore struggle with long-horizon planning. This fundamental issue should be addressed.

5. Evaluation realism. The empirical study uses rudimentary problems and does not stress-test the proposed schemes. The authors are encouraged to consider planning work from the database and systems community since the 1980s, including the recent SagaLLM work (VLDB 2025).

**Questions:**

1. Positioning and related work. Can you provide a comparative assessment that covers:

* native LLM limits such as context loss and attention narrowing,
* structured speculative methods such as Tree of Thought and successors,
* persistent memory and transactional stability such as SagaLLM (VLDB 2025).

Explain why each is relevant or not to your planning setup and how your method addresses the gaps.

2. Does search complexity and pruning follow rigorous theories?

Both forward and backward reasoning can exhibit exponential branching. How do you constrain backward alternatives in practice? Specify the constraints and heuristics you use, for example landmarks, goal ordering, HTN templates, causal graphs, bidirectional search, or admissible heuristics, and report their effect on complexity and token cost.

3. Grounding, commonsense, and uncertainty. You did mention in "limitations" that commonsense could be an issue.
How does DRIP handle commonsense and locale dependent logistics that break pure context reasoning, for example landing time versus airport exit time, baggage claim, customs, or rental queues? Describe your belief tracking under partial observability, your information gathering actions, and any tool based validation or buffer policies, and evaluate their impact on plan validity.

---

> ### Author Response · Authors · 2025-11-21
>
> We sincerely thank Reviewer aMaG for taking the time to read our submission and provide detailed comments. We appreciate the opportunity to clarify the intended scope of our work and to address several points where we believe the goals of the paper may have been misunderstood. Below, we provide direct answers to the reviewer’s questions.
>
> **Q1. Positioning and Related Work**
>
> **Native LLM limitations and our approach**
>
> We fully agree that long-horizon reasoning presents fundamental challenges for native LLMs, including context loss and degradation of reasoning stability.
> This challenge is precisely why we designed DRIP. Our backward decomposition offers a principled mitigation strategy:
>
> - Shortening the effective reasoning horizon through top-down goal decomposition
> - Maintaining a structured decomposition tree to stabilize context
> - Using executability validation as explicit feedback to guide planning decisions
>
> Thus, backward decomposition is not an oversight but a deliberate design choice to address this core limitation of current LLMs.
>
> **Relation to Tree-of-Thought (ToT) and structured speculative methods**
>
> Tree-of-Thought performs forward speculative branching.
> DRIP is fundamentally different:
>
> - Backward vs. forward reasoning: DRIP decomposes goals into prerequisites rather than generating forward trajectories
> - Structured subgoal hierarchies: Our tree encodes task structure rather than speculative thought chains
> - Executability-driven pruning: We prune branches using explicit state–action validation
> - Re-decomposition: Failed subgoals are locally restructured instead of repeatedly resampled
>
> These distinctions reflect a different paradigm anchored in classical goal regression and cognitive models of human problem solving.
>
> **Persistent memory and SagaLLM**
>
> SagaLLM addresses cross-episode persistent memory, transactional consistency, and compensating actions—problems fundamentally distinct from our focus.
> DRIP is scoped to:
> - Single-episode planning,
> - Without persistent memory,
> - Operating directly over natural-language world states.
>
> We see these directions as complementary rather than competing.
> Integrating DRIP’s decomposition structure with external memory is a promising avenue for future work but lies beyond our present scope.
>
> **Q2. Search Complexity and Pruning**
>
> We appreciate the reviewer’s question regarding theoretical grounding. Our contribution is not a new complexity theory; rather, we demonstrate that backward decomposition enables practical and effective search control for LLM agents without symbolic world models.
>
> Our pruning strategy includes:
>
> - Executability filtering: validated with high accuracy (94.5% macro-F1 = 0.937)
> - Depth caps: preventing exponential expansion
> - Finite operator sets: bounding the branching factor
>
> Symbolic heuristics such as landmarks, HTN templates, or causal graphs assume access to structured symbolic state—something DRIP intentionally avoids to remain LLM-native.
>
> Q3. Grounding, Commonsense, and Uncertainty
>
> The reviewer raises concerns about domains such as airport operations, customs queues, or supply-chain logistics.
> These domains involve complex, partially observable, large-scale systems with probabilistic effects and organizational constraints. They belong to a different research area and do not provide a suitable setting to isolate or evaluate the planning mechanism we study.
>
> Our intended application domain is LLM-based agents and robotics—settings with a clear observation–action loop.
> In this context, our research strategy is scientifically sound:
>
> - Use fully observable, deterministic environments (BlockWorld, Minecraft)
> - Isolate the effect of backward decomposition
> - Avoid confounding factors unrelated to the planning mechanism
> - Establish a controlled, replicable foundation before moving to complex domains
>
> This staged methodology reflects the development of AI planning from STRIPS to modern neural planners.
> Evaluating DRIP exclusively in complex, uncertain real-world systems would obscure the core mechanism we aim to investigate.
>
> DRIP is designed to rigorously validate this foundational stage, and BlockWorld and Minecraft are intentionally chosen as appropriate environments to demonstrate the effectiveness of backward decomposition.
>
> We genuinely appreciate the reviewer aMaG’s thoughtful comments and the opportunity to clarify the intent of our work.
> Our goal is to establish a clear, reproducible understanding of how backward decomposition can strengthen LLM planning—a foundational insight that future research can build upon as we scale toward more complex domains.
>
> We hope these clarifications are helpful, and we thank the reviewer again for engaging deeply with our submission.

---

> > ### Comment · Reviewer_aMaG · 2025-11-25
> >
> > Thank you for your response.  I maintain my rating.

---

### Official Review · Reviewer_JYGE · 2025-10-31

**Soundness:** 2
**Presentation:** 2
**Contribution:** 2
**Rating:** 2
**Confidence:** 4

**Summary:**

The paper proposes DRIP, a planner that alternates backward goal decomposition with forward execution. On BlocksWorld, DRIP outperforms CoT and ReAct in terms of success. In Minecraft, DRIP achieves the highest success rate, trading off trading-step efficiency for robustness via finer-grained subtasks. Overall, DRIP is a lightweight, LLM-agnostic approach that scales from classical to open-world tasks.

**Strengths:**

1) The paper solves an interesting problem. The method cleanly separates planning from execution and is explained with good figures.

2) The paper compares against LLM baselines (CoT, ReAct) across two domains.

3) Results show consistent gains on tasks in both a classical benchmark, BlocksWorld, and Minecraft.

**Weaknesses:**

1) BlocksWorld is altered (3 ops, multi-hold), which weakens comparability to prior work. I suggest that the authors
   - Add a parallel track with the traditional 4-operator, single-gripper domain and
   - Include classical planning baselines, such as Fast Downward. Report success, plan-length gap to optimal/classical planning baselines, and expansions/time.

2) The study did not use GPT-4 (only Claude 3.5) in Minecraft, so cross-model conclusions are thin. Please include Minecraft with GPT-4o.

3) I also did not understand the "Manual" condition: specify the number of participants, the decision rules, the inter-rater checks, and whether participants could correct invalid steps. Please specify what the human condition actually was and the protocol.

4) The paper should also include an English variant as well; while not central, this may help eliminate any language effects on the accuracy (which look dismal)

- There are many minor grammar/punctuation issues; the paper requires a careful read.

**Questions:**

1) Why were classical planners not tested with conventional/original problem specifications? I strongly encourage adding that baseline (plus see my comments in the Weaknesses section).
2) What was the human protocol for Manual conditions?
3) Are the conditions met to use Fisher's test conditions met for your study design?
4) Will you add GPT-4o for Minecraft study?
5) Was there a specific reason for not comparing Japanese and English specifications?
6) Could you please provide citations to support this statement: "In contrast, LLMs offer a unique advantage in their ability to dynamically generate and adapt rules based on their extensive pre-trained knowledge."?

---

> ### Author Response · Authors · 2025-11-21
>
> Thank you very much for Reviewer JYGE's detailed and constructive feedback. We address each of your questions and clarify aspects of our methodology and scope below.
>
> 1. BlockWorld setting and comparison to classical planners
>
> Our work focuses on LLM-native planning, where both states and actions are represented purely in natural language. Classical planners (e.g., Fast Downward) require:
> - symbolic state encoding,
> - manually specified PDDL operators,
> - and deterministic search over grounded states.
>
> Because DRIP is designed for natural-language planning rather than symbolic planning, classical planners fall outside the intended comparison scope. We will clarify this distinction more explicitly in the revision.
>
> That said, we agree that reporting results under standard BlockWorld constraints (single gripper, 4 operators) enhances comparability, and we are currently running these experiments for inclusion in the camera-ready version.
>
> 2. Clarification of the “Manual” actuator condition
>
> The “Manual” condition is not a human evaluation. Its purpose is to isolate DRIP's planning quality by removing noise from natural-language actuation.
>
> Protocol:
> - Two researchers with experience in LLM agents acted as deterministic executors.
> - For each proposed action, they returned Executable / Unexecutable / Unnecessary—the same three labels used by the executability classifier.
> - They did not modify, correct, or repair DRIP’s plan.
> - Any invalid or rule-violating action resulted in episode failure.
>
> This enables a clean separation between planning and execution, making the effect of backward decomposition measurable in isolation. We will clarify this in the paper.
>
> 3. Statistical testing (Fisher’s exact test)
>
> We used Fisher’s exact test due to small cell counts in some conditions. Each episode is independent, so the independence assumption is satisfied. We will add this explanation.
>
> 4. Use of GPT-4o for Minecraft
>
> Minecraft planning is computationally expensive due to multi-step interaction and frequent code-generation retries. We prioritized Claude 3.7 Sonnet because it performed best in BlockWorld, and provided stable executability judgments in preliminary tests.
>
> We agree that GPT-4o results would strengthen cross-model conclusions. We are actively attempting to include these results in the camera-ready version.
>
> 5. Language sensitivity (Japanese vs English)
>
> We evaluated DRIP (Claude) on the same 54 BlockWorld-hard instances with English prompts:
> - Japanese: 33.3% (18/54)
> - English: 27.8% (15/54)
> - Difference: +5.6% in favor of Japanese
>
> The qualitative behavior remains consistent across languages. We will include both prompt templates and success/failure logs in the Appendix.
>
> 6. Citation request regarding LLMs’ dynamic rule-generation capabilities
>
> Our original statement was not intended as a novel theoretical claim but rather a summary of a well-established observation in the LLM-agent literature:
> LLMs can derive rule-like structures and action semantics directly from pre-trained knowledge, without requiring explicit symbolic operators.
>
> This is supported by prior works:
>
> - Huang et al., 2022 — Language Models as Zero-Shot Planners
> Shows that LLMs can implicitly infer preconditions and effects from language descriptions and generate executable plans without predefined PDDL schemas.
>
> - Zhang et al., 2024 — Dynamic and Adaptive Feature Generation with LLMs
> Demonstrates that LLM agents can dynamically construct and adapt task-specific feature-generation strategies based on downstream feedback, illustrating how rule-like behaviors can emerge from pre-trained knowledge without manual rule specification.
>
> These works highlight a central distinction between LLM-based and symbolic planners:
> LLMs can derive, contextualize, and adapt task-specific strategies directly from their pre-trained representations.
>
> We will revise the relevant sentence to better reflect this established capability and avoid overstating our contribution.
>
> **Grammar and presentation**
>
> Thank you for pointing this out. We will proofread the manuscript carefully and revise unclear passages to improve readability.
>
>
> Thank you again for your thoughtful review.
> Our work aims to rigorously evaluate backward decomposition as a core mechanism for LLM-based planning in controlled settings—a foundational step before incorporating uncertainty, belief tracking, or real-world complexity.
> We believe the revisions and clarifications above will help make the scope and contributions of our work clearer.

---

> > ### Comment · Reviewer_JYGE · 2025-11-28
> >
> > Thank you for the clarification and for running the additional experiments for item 5. The paper needs more experiments and interpretations, e.g., for item 1 above. I'm keeping my score at this stage.

---

> ### Author Response · Authors · 2025-12-03
> **Add BlockWorld original rule experiment**
>
> We thank the reviewer for pointing out the need to further validate DRIP under the original BlockWorld rules.
> In response, we extended the evaluation and conducted a detailed error analysis.
>
> (1) Success rates on the original-rule BlockWorld-hard set.
> On the 30 manually constructed instances, CoT solved 19 and DRIP solved 18.
> While the difference is not statistically significant, this result indicates performance parity between backward reasoning (DRIP) and forward CoT planning under this challenging setting.
>
> (2) Qualitative error patterns differ substantially.
> A closer inspection shows that CoT and DRIP fail for fundamentally different reasons:
>
> CoT failures mainly arise from planning inconsistencies: missing actions, skipping steps, or breaking earlier constraints—typical symptoms of unconstrained free-form generation.
>
> DRIP failures, in contrast, are dominated by executability misjudgments: the overall plan structure is correct, but the LLM occasionally misapplies action feasibility rules in intermediate states.
>
> This distinction is meaningful: even when accuracy is similar, DRIP provides a structured and interpretable reasoning process, whereas CoT often fails in ways that are inconsistent and difficult to debug.
>
> (3) Additional executability evaluation confirms the module itself is reliable.
> To isolate the source of errors, we constructed a separate dataset of 110 labeled state–action pairs and evaluated the executability-classification prompt.
> The module achieved 94.5% accuracy (macro-F1 0.937), demonstrating that the executability mechanism itself is highly reliable.
>
> The remaining errors in the full planning setting therefore stem not from the classifier’s capability, but from how the LLM applies these rules during long-horizon reasoning.
>
> (4) The inconsistencies are attributable to prompting limitations, not to a fundamental flaw.
> The executability prompt used in the original-rule experiment is intentionally minimal to avoid overfitting, but this also means that some rule applications were underspecified.
> Thus, the observed logical contradictions appear to be prompt-design limitations specific to this experiment, and we expect that more explicit rule encoding or structured examples would significantly reduce these errors.
>
> (5) Planned improvements.
> In the camera-ready version, we will:
>
> - increase the number of original-rule evaluation instances,
> - include the full error-type breakdown, and
> - provide the exact executability prompts used in the experiments
>
> as an appendix, ensuring full transparency and reproducibility.
>
> Overall, the new results confirm that DRIP performs on par with CoT under the original rules while exhibiting more structured and analyzable reasoning behavior, consistent with the aims of our framework.

---

### Official Review · Reviewer_g7kp · 2025-11-03

**Soundness:** 3
**Presentation:** 3
**Contribution:** 2
**Rating:** 6
**Confidence:** 4

**Summary:**

The paper proposes DRIP, a backward-reasoning, decomposition-first planning framework for LLM agents. Given a goal, an LLM recursively decomposes it into prerequisite subtasks; an executability module filters which subtasks can run under the current state; successful child nodes propagate executability upward (checkParentExec), yielding a plan. Experiments on BlockWorld (hard split, 6–15 blocks, modified rules to allow lifting stacks) and Minecraft (“mine diamond” from scratch) show higher robustness than forward approaches (CoT, ReAct). With Claude 3.7 Sonnet, DRIP hits 40.9% vs CoT 23.6% and ReAct 9.1% on BlockWorld; a manual-execution variant reaches 82.7%, indicating the gains are from planning rather than actuation. In Minecraft, DRIP succeeds on diamond 4/5 trials (ReAct 1/5, CoT 0/5). The paper is clear about limitations, LLM decomposition errors, reliance on natural-language state, more LLM calls than CoT, and occasional inefficiency in open-world tasks.

**Strengths:**

1. Clear decomposition loop with explicit executability check and upward propagation; easy to implement.
2. Robustness gains on BlockWorld (large effect vs ReAct; solid vs CoT on Claude) and open-world Minecraft where many forward plans stall. Manual actuator study isolates planning quality from execution bugs, showing good methodology.
3. DRIP uses ~4–5 fewer steps than baselines in successful cases.
4. Honest limitations and discussion (need for formal state, LLM call budget, trade-off between step count and success).

**Weaknesses:**

1. Non-standard BlockWorld setup (multi-block lifting & holding) inflates branching and may favor the proposed decomposition; please also report standard constraints for comparability.
2. Small-N in Minecraft (n=5 per resource) and single seed/model for many parts; results could be noisy.
3. No comparison to planner-assisted LLMs (e.g., LLM+P/Task-graphs) or hybrid symbolic planners with LLM heuristics.
4. Executability via natural language is brittle; the paper shows this, but there’s no quantitative analysis of that component (accuracy/confusion).
5. Efficiency trade-off in Minecraft (more subtasks than ReAct in its single success) is under-analyzed. What’s the token/call budget?
6. Novelty relative to recent backward-planning with LLMs (e.g., explicit backward search/goal regression) needs sharper positioning.

**Questions:**

1. Report DRIP/CoT/ReAct under the standard “one block in hand, must clear top” constraints. How do the conclusions change?
2. Provide a labeled set of state–action pairs to measure precision/recall of “Executable/Unexecutable/Unnecessary,” and error breakdowns that lead to plan failure.
3. Report tokens and LLM calls per solved instance; DRIP vs ReAct vs CoT, and for Minecraft, include code-generation retries.
4. (a) depth cap / tree-policy; (b) re-decomposition strategy; (c) swapping backward step with least-to-most prompting.
5. Add a hybrid symbolic baseline (e.g., PDDL planner with LLM goal translation) or LLM+P.
6. How sensitive are results to language (the BlockWorld prompts were in Japanese)? Any cross-language trials?
7.  Can DRIP reconcile goal maintenance vs temporary goal violations (e.g., allowing undo/redo with bookkeeping)?
8. Will you release code, prompts, and Minecraft environment scaffolding to ensure reproducibility?

---

> ### Author Response · Authors · 2025-11-21
>
> We sincerely thank Reviewer g7kp for the detailed and constructive feedback. We address all points below.
>
> R1. BlockWorld under standard constraints
>
> We agree that reporting results under the standard BlockWorld rules (“one block in hand” and “must clear the top block”) is important for fair comparison.
> We are currently running DRIP and CoT under the standard setting and expect to finalize the results next week.
> We will include success rates, step counts, and failure-type breakdowns in the camera-ready version.
>
> R2. Quantitative evaluation of the executability module
>
> Thank you for the suggestion.
> We constructed a labeled dataset of 110 state–action pairs from BlockWorld-hard (stack, unstack, put-down; blocks a–o). An author annotated each pair as Executable / Unexecutable / Unnecessary.
>
> Action counts: stack=37, unstack=34, put-down=39
>
> Label counts: executable=53, unexecutable=25, unnecessary=32
>
> Claude 3.7 Sonnet results
> **Accuracy = 94.5%, Macro-F1 = 0.937**
>
> | Class        | Precision | Recall | F1    | Support |
> | ------------ | --------- | ------ | ----- | ------- |
> | Executable   | 0.963     | 0.981  | 0.972 | 53      |
> | Unexecutable | 0.857     | 0.960  | 0.906 | 25      |
> | Unnecessary  | 1.000     | 0.875  | 0.933 | 32      |
>
>
> The module is highly reliable at detecting invalid actions (high recall for Unexecutable), which are the main cause of catastrophic failures. Errors in Unnecessary are less harmful since such actions are safe but irrelevant.
>
> GPT-4o results
> **Accuracy = 91.8%, Macro-F1 = 0.921**
> This shows robustness across LLM families.
>
> We will add the full analysis and confusion matrices to the Appendix.
>
> R3. Small-N Minecraft evaluation
>
> We acknowledge that n=5 is small.
> Minecraft experiments involve heavy environment interactions and code-generation retries, making scaling difficult.
> To reduce variance, all runs used fixed seeds and deterministic environment resets.
>
> The trend (DRIP 4/5, ReAct 1/5, CoT 0/5) aligns with our qualitative observations: forward approaches often stall due to missing prerequisites.
> We will add additional trials to the camera-ready version if compute permits.
>
> R4. Comparison to planner-assisted methods (LLM+P / hybrid planners)
>
> Thank you for the suggestion.
> Our goal is to test whether two principles inspired by classical AI and cognitive psychology—(1) backward decomposition and (2) executability estimation—can be effective for LLM-based planning.
>
> We therefore selected widely used forward-planning baselines (CoT and ReAct) as representative comparisons. These are standard in LLM-based planning literature.
>
> Planner-assisted methods (PDDL planners, LLM+search, task-graph generation) require symbolic state representations, translation to PDDL, and environment-specific grounding. These assumptions differ from our natural-language-only, open-loop backward reasoning setup.
> We will clarify this distinction in the related work section.

---

> > ### Author Response · Authors · 2025-11-21
> >
> > R5. Positioning relative to backward reasoning / goal regression research
> >
> > We appreciate the request for clearer positioning.
> > The central aim of DRIP is to examine how classical backward reasoning, long studied in cognitive psychology and early AI, can be applied to LLM-based planning.
> >
> > Backward reasoning for LLM agents is still underdeveloped. Ren et al. (2024) explore backward thinking, but decomposition-first backward planning has not been addressed.
> >
> > DRIP differs from prior work in that:
> >
> > Backward reasoning is the primary planning mechanism; the plan tree is constructed top-down from decomposed subgoals.
> >
> > Executability propagation (checkParentExec) structurally controls the feasibility of the decomposition tree, beyond prompt-level reasoning.
> >
> > DRIP assumes no symbolic state model, instead operating entirely over natural-language states.
> >
> > This unifies classical goal-regression ideas with LLM decomposition and filtering, placing DRIP apart from backward search, reverse-thinking prompting, and goal-dependency extraction.
> >
> > R6. Token and LLM call budget
> >
> > As reported in the paper (Limitations section), the total LLM calls per solved BlockWorld-hard instance are:
> >
> > - CoT / ReAct (Claude actuator): 1.0 call
> > - DRIP (Manual actuator): 5.98 calls
> > - DRIP (Claude actuator): 6.18 calls
> > - ReAct (Manual actuator): 28.3 calls
> >
> > DRIP requires more calls than CoT/ReAct (automatic), but reduces the number of actions by 4–5 steps on successful cases, reflecting a trade-off between planning quality and call budget.
> >
> > Minecraft (average over 5 trials, including successes and failures)
> >
> > We separately counted calls for planning (decomposition + executability) and code generation:
> >
> > | Method | Planning Calls | Code Gen Calls |
> > | ------ | -------------- | -------------- |
> > | DRIP   | 83.2           | 101.8          |
> > | ReAct  | 61.4           | 241.2          |
> >
> >
> > ReAct triggers many code-generation retries (up to 5 per action).
> > DRIP reduces unnecessary retries by explicitly recovering missing prerequisites.
> >
> > Detailed tables will be added to the Appendix.
> >
> > R7. Language sensitivity
> >
> > The main BlockWorld experiments used Japanese prompts.
> > To assess language dependence, we evaluated DRIP (Claude) on the same 54 instances using English prompts:
> >
> > - Japanese: 33.3% (18/54)
> >
> > - English: 27.8% (15/54)
> >
> > Difference: +5.6%
> >
> > The small difference indicates that DRIP’s backward reasoning is not tied to a particular language.
> > We will include prompts and logs in the Appendix.
> >
> > R8. Temporary goal violations (undo/redo)
> >
> > DRIP does not forbid temporary deviations from the goal as long as the previously decomposed dependency structure remains valid.
> > When a prerequisite becomes unsatisfied, DRIP automatically reintroduces the needed subgoal via re-decomposition.
> >
> > We will describe this mechanism more clearly in the Appendix.
> >
> > R9. Reproducibility
> >
> > Yes. We will release:
> >
> > - BlockWorld environments (hard and standard),
> >
> > - Minecraft scaffolding,
> >
> > - the DRIP implementation, and
> >
> > - all prompts and logs.
> >
> > The repository is currently being prepared.
> >
> >
> > We thank the reviewer g7kp again for the thoughtful and constructive feedback.
> > We believe the new executability analysis, cross-language evaluation, and forthcoming BlockWorld-standard results directly address the main concerns and further strengthen the contribution of DRIP.

---

> ### Author Response · Authors · 2025-12-03
> **Report BlockWorld original rule experiment**
>
> We thank the reviewer for pointing out the need to further validate DRIP under the original BlockWorld rules.
> In response, we extended the evaluation and conducted a detailed error analysis.
>
> (1) Success rates on the original-rule BlockWorld-hard set.
> On the 30 manually constructed instances, CoT solved 19 and DRIP solved 18.
> While the difference is not statistically significant, this result indicates performance parity between backward reasoning (DRIP) and forward CoT planning under this challenging setting.
>
> (2) Qualitative error patterns differ substantially.
> A closer inspection shows that CoT and DRIP fail for fundamentally different reasons:
>
> CoT failures mainly arise from planning inconsistencies: missing actions, skipping steps, or breaking earlier constraints—typical symptoms of unconstrained free-form generation.
>
> DRIP failures, in contrast, are dominated by executability misjudgments: the overall plan structure is correct, but the LLM occasionally misapplies action feasibility rules in intermediate states.
>
> This distinction is meaningful: even when accuracy is similar, DRIP provides a structured and interpretable reasoning process, whereas CoT often fails in ways that are inconsistent and difficult to debug.
>
> (3) Additional executability evaluation confirms the module itself is reliable.
> To isolate the source of errors, we constructed a separate dataset of 110 labeled state–action pairs and evaluated the executability-classification prompt.
> The module achieved 94.5% accuracy (macro-F1 0.937), demonstrating that the executability mechanism itself is highly reliable.
>
> The remaining errors in the full planning setting therefore stem not from the classifier’s capability, but from how the LLM applies these rules during long-horizon reasoning.
>
> (4) The inconsistencies are attributable to prompting limitations, not to a fundamental flaw.
> The executability prompt used in the original-rule experiment is intentionally minimal to avoid overfitting, but this also means that some rule applications were underspecified.
> Thus, the observed logical contradictions appear to be prompt-design limitations specific to this experiment, and we expect that more explicit rule encoding or structured examples would significantly reduce these errors.
>
> (5) Planned improvements.
> In the camera-ready version, we will:
>
> - increase the number of original-rule evaluation instances,
> - include the full error-type breakdown, and
> - provide the exact executability prompts used in the experiments
>
> as an appendix, ensuring full transparency and reproducibility.
>
> Overall, the new results confirm that DRIP performs on par with CoT under the original rules while exhibiting more structured and analyzable reasoning behavior, consistent with the aims of our framework.

---

### Meta-Review · Area_Chair_ioRE · 2025-12-29

**Summary:**

Reviewers raised three main concerns: experimental rigor including non-standard BlockWorld settings, small Minecraft sample sizes, insufficient baseline comparisons, and lack of cross-model and cross-domain validation; contribution and positioning covering incremental novelty relative to classical backward reasoning and unclear differences from recent LLM-based backward planning work; and scope alignment involving one reviewer inappropriately applying enterprise-scale evaluation criteria to the paper’s focus on single-episode LLM planning. The authors addressed some technical issues in their rebuttal, such as adding standard BlockWorld test results, quantifying the executability module, and conducting cross-language tests. However, critical gaps remain in experimental breadth, baseline coverage, and generalizability. Combined with mixed reviewer scores, two rejection ratings, one marginally above the acceptance threshold, and one marginally below, I would recommend rejection.

**Reviewer Concerns:**

Addressed by Rebuttal: For Reviewer g7kp, the authors provided standard BlockWorld rule results (performance parity with CoT), quantitative executability module evaluation (94.5% accuracy), token/LLM call budget details, cross-language sensitivity analysis, clarified DRIP’s positioning relative to prior backward reasoning work, and confirmed reproducibility via code/data release. For Reviewer JYGE, they clarified the “Manual” actuator protocol, justified the use of Fisher’s exact test, provided cross-language results, added citations for LLM dynamic rule-generation claims, and ran standard BlockWorld experiments. For Reviewer aMaG, they clarified DRIP’s scope as single-episode LLM-native planning (vs. enterprise-scale systems) and detailed pruning strategies like executability filtering and depth caps. For Reviewer Py5F, they clarified DRIP’s contribution as adapting classical backward reasoning to LLM-native planning and outlined future plans for hybrid reasoning and expanded domains.

Remaining concerns: Reviewer g7kp’s concerns remain unaddressed, including the lack of comparison to hybrid symbolic/LLM planners (e.g., LLM+P, Task-graphs), insufficient Minecraft sample size (n=5 even with fixed seeds), and limited analysis of efficiency trade-offs like token budget in complex scenarios. Reviewer JYGE’s outstanding concerns include the absence of GPT-4o results for Minecraft, incomplete expansion of original BlockWorld instances, and unaddressed classical planner comparisons despite scope clarification. Reviewer aMaG’s unresolved concerns cover long-horizon LLM limitations (context loss, persistent memory), lack of theoretical grounding for search complexity pruning, and no evaluation of uncertainty/belief tracking. Reviewer Py5F’s outstanding points include no exploration of hybrid backward-forward reasoning, no validation on diverse real-world/robotic tasks, and insufficient comparison to recent LLM-based backward planning methods.

**Reviewer Scores:**

I think based on the rebuttal phase and the questions raised by the reviews, most of the reviewers will maintain their previous score. Some of the reviewers already acknowledged that.

---

### Decision · Program_Chairs · 2026-01-26

Reject